# University students' health-related quality of life and its determinants. Results from a cross-sectional survey during the COVID-19 pandemic

Karoline Wagner[1], Zoë Reinhardt[2], Sarah Negash[1], Lena Weber[1], Andreas Wienke[1], Rafael Mikolajczyk[1,3,4], Amand Führer[1]*

1 Institute for Medical Epidemiology, Biometrics and Informatics (IMEBI), Interdisciplinary Center for Health Sciences, Medical School of the Martin Luther University Halle-Wittenberg, Halle (Saale), Germany, 2 Department of Education and Pedagogy, Faculty of Philosophy III, Martin Luther University Halle-Wittenberg, Halle (Saale), Germany, 3 German Center for Mental Health (DZPG), Site Jena-Magdeburg-Halle, Halle (Saale), Germany, 4 Center for Intervention and Research on Adaptive and Maladaptive Brain Circuits Underlying Mental Health (C-I-R-C), Jena-Magdeburg-Halle, Halle (Saale), Germany

* amand-gabriel.fuehrer@uk-halle.de

**Editor:** D. Daniel, Gadjah Mada University Faculty of Medicine, Public Health, and Nursing: Universitas Gadjah Mada Fakultas Kedokteran Kesehatan Masyarakat dan Keperawatan, INDONESIA

## Abstract

Being a university student is a vulnerable period marked by transitions and uncertainties which can impair their physical and mental well-being as well as overall quality of life. The existing literature suggests that certain groups of students might be particularly affected by that. In addition, quality of life might have been further affected by the COVID-19 pandemic. Therefore, this article investigated university students' quality of life and its determinants during the COVID-19 pandemic using an online survey among students of one German university. Quality of life was measured using SF-12's physical (PCS) and mental dimension (MCS). We conducted stratified descriptive analyses followed by regression analyses. 875 respondents completed the questionnaire, of whom 63.0% were female and 95.7% had a German nationality and 16.0% reported having a migration background. Mean age was 23 years. Medical (17.7%) and law students (15.8%) were the biggest groups, but all other faculties of the university were also covered. Concerning respondents' health-related quality of life, mean PCS was 51 (95%CI = (47–55)), while mean MCS was 36 (95%CI = (30–43)). Students with non-German nationality (PCS: 49; MCS: 29) and students with migration background (PCS: 50; MCS: 31) showed particularly low quality of life in the mental dimension. In multivariable regression analyses, associations of the included determinants with PCS were rather weak. In contrast, there were strong associations between MCS and having a migration background with regression coefficient β = -26.1 (95%CI = (-38.5 – -13.7)) and studying Law with β = -17.7 (95%CI = (-28.2 – -7.2)). When comparing these findings with pre-pandemic studies, it seems that university students' quality of life during the pandemic was as low as it had been before while–as in pre-pandemic studies–certain groups of students generally fared worse than others did. This suggests that universities should actively promote students' health and should implement measures to specifically support

**Data Availability Statement:** The underlying dataset contains a number of sociodemographic characteristics (such as age, gender, subject, nationality, etc.) that might allow a de-anonymization of participants in a public dataset. Therefore, the ethics clearance for this study qualified that data can only be published "in aggregated form". As a result, data cannot be made publicly available but will be available from the responsible Institutional Review Board (email: ethik-kommission@uk-halle.de) upon reasonable request.

**Funding:** This research was supported by a grant from Amadeu Antonio Foundation (grant number 21/003).

**Competing interests:** The authors have declared that no competing interests exist.

particularly vulnerable students such as students with migration background or students of certain subjects.

## Introduction

Being a university student is a time of great potential, but can also be fraught with uncertainties and upheavals. Life-course epidemiology has long established the time between 12 and 24 years of age as an important transition period from childhood to adulthood: This time comes with the challenge of making career choices, moving from education to working life, and evolving from adolescents' social roles towards adult responsibilities [1]. For university students, this transition is additionally charged with a number of vulnerabilities [2, 3] that are connected to stressors such as financial insecurity [4, 5], uncertainties about the future [6], and academic pressure [7, 8].

These stressors have the potential to impair students' quality of life and affect their health [2, 9], which in turn can influence their academic success and career prospects [3]. Therefore, in recent years, students' health and well-being has been the focus of an increasing amount of research [3]. This research has focused on different indicators of student health, from self-rated health [10] and mental health [11] to sleep quality [12], psychosocial well-being, and health-related quality of life [3]. In this context, quality of health is understood to be a multidimensional concept that includes social, environmental, social and mental dimensions and can be used as a surrogate parameter for overall wellbeing while also being closely associated with a vast variety of mental and physical health outcomes [13, 14].

The above-mentioned studies found alarmingly poor mental health among university students [15, 16], with prevalences of depression and anxiety much higher than in the general population of the same age [17, 18]. Similarly, students' self-rated health and health-related quality of life are often substantially lower than in their non-student peers [19].

Considering the fact that–overall–university students are a relatively young demographic group, these comparatively poor outcomes in different health indicators may seem surprising. Accordingly, a number of studies investigated potential factors affecting university students' health and found that subjective social status and financial situation, social network, personality traits, and students' gender are associated with mental health status and quality of life. An Australian study found students' mental health to be strongly associated with their subjective social status, with the frequency of social contact with university friends as an important mediator of this relationship [20]. Beyond subjective measures, having debt because of student loans [4] and lacking sufficient financial means for every-day life [5, 9] have been found to increase mental health issues among students and impair their health-related quality of life. Similarly, feeling unable or overwhelmed by the university's academic requirements leads to stress and reduces quality of life [7], as does the feeling of lacking agency concerning the planning of their time at university [8, 21].

When it comes to individual factors, female gender and studying in the first semesters of university have been shown to be associated with higher prevalences of symptoms of anxiety and depression [6, 10, 16, 17], while individual personality traits such as optimism were shown to be associated with better mental health [22]. Also, the experience of discrimination has been found to reduce students' quality of life, both if the discrimination occurs outside of university and within the university context [23].

In addition to those stressors, the effects of the COVID-19 pandemic have been shown to further impair university students' quality of life and exacerbate preexisting mental health issues [24–26]. Hereby studies have shown that the onset of the pandemic and the resulting changes in university routines (such as online teaching) increased anxiety and depressive mood, were perceived as stressful, and resulted in a thinning of students' social nets [27, 28]. In addition, studies suggest that students from marginalized groups were affected more strongly by this issue [29, 30].

Still, as of now, only a few studies analyze which determinants of students' mental health were important during the pandemic. This knowledge gap is problematic since under the conditions of the pandemic, different resilience and risk factors might become important compared to the pre-pandemic situation [11, 25, 26]. Considering this background of research, the aim of this study was to measure students' health-related quality of life and to assess which factors are associated with quality of life in this population under the conditions brought forth by the COVID-19 pandemic. In order to further the current state of knowledge, we hereby collected information on a wide range of potential variables associated with quality of health.

## Materials and methods

This study is based on the findings of a cross-sectional online survey, which was conducted among university students as part of a study on students' experience of discrimination at a German university. Data was collected between 18/03/2021 and 31/05/2021 using the survey-software Limesurvey.

### Sampling

Participants were recruited among the students of the Martin Luther University Halle-Wittenberg using convenience sampling. We aimed to include all students enrolled at this university at the time of the survey and used a number of mailing lists kept by different university departments and student organizations to distribute the invitations. In addition, we advertised the study on the university's online learning platform, and a number of counselling offices shared the invitation on social media. Since we have no way to establish how many students were reached through these channels, we are unable to calculate the response proportion.

### Questionnaire

For the purpose of this study, the questionnaire collected three types of data: First, it measured respondents' health-related quality of life using the German version of the SF-12 [31]. The choice of the SF-12 was motivated by its widespread use and the resulting option to directly compare our findings among university students during the pandemic with previous research in this target group before the pandemic [3] but also with other, non-student populations [32, 33].

Second, it gathered sociodemographic information such as age, gender, nationality and migration background (the latter defined as having at least one parent with non-German nationality), and information on the respondents' university studies. Third, it assessed a number of potential stressors that might impair students' quality of life. To that end, respondents were asked whether they considered themselves to be members of a group (or groups) of people potentially affected by discrimination. If they responded in the affirmative, they were asked to specify the group it in an open question. In addition, they were asked if they had actually experienced discrimination in the context of university life. We also collected data on whether participants had children, were affected by a chronic illness or disability, and whether they considered their financial means to be sufficient.

## Statistical analyses

The SF-12 was evaluated using standard procedures [31] for separately calculating the Mental Component Summary (MCS) and the Physical Component Summary (PCS), whereby each score can reach values between 0 and 100, and higher values signal better health-related quality of life [34]. We report the average MCS and PCS values and their 95% confidence intervals. To assess potential determinants of quality of life, we performed stratified analyses in which we compared MCS and PCS across different sociodemographic groups.

In a last analytic step, we aimed to account for multiple variables associated with quality of life and accordingly performed linear regression with the physical and the mental dimension of quality of life as dependent variables and the different sociodemographic characteristics and stressors as independent variables. Hereby, we first conducted simple regression analyses; then in a second step, we calculated one multivariable model for each dependent variable, in which the majority of independent variables were included. We did not include nationality and membership of a potentially discriminated group in order to avoid collinearity with migration background, gender, religion, nationality and experience of discrimination. We report regression coefficients and their respective 95% confidence intervals.

## Ethics

The Institutional Review Board of Martin Luther University Halle-Wittenberg granted ethical approval for the study (registration number: 2020–144). Data collection was anonymous and in accordance with the respective data protection laws. Before starting the survey, respondents were informed about the aim of the study, anonymity, and measures of data protection and gave their consent to participate by clicking the respective field.

## Results

Among all 1672 respondents, 875 completed the entire questionnaire and were thus included in this analysis. Respondents had a mean age of 23 years. 63.0% (n = 514) were female, 95.7% (n = 837) were of German nationality and 16.0% (n = 142) reported having a migration background. The most frequently reported university subjects were Medicine (17.7%, n = 142) and Law (15.8%, n = 138), followed by Educational Science (7.0%, n = 61), High School Teaching (6.0%, n = 53) and Sociology (3.8%, n = 33). Most respondents received financial support from their parents (66.0%, n = 579) and 4.8% (n = 42) reported having underage children themselves. More details on sociodemographic characteristics are displayed in Table 1.

When asked about stressors potentially impairing their well-being, 40.0% of all respondents reported membership of at least one group potentially affected by discrimination, of which *women* (n = 203) was the most commonly reported group, followed by *LGBTQ+* (n = 92), *migrants* (n = 52) and *people of color* (n = 39). 29.0% reported that they actually had experienced discrimination within the context of university. Further details on potential stressors are shown in Table 2.

## Quality of life in different groups of students

Overall, the average PCS was 51.2 and the MCS was 36.3. Female students showed a slightly lower MCS than male students, while PCS was very similar. German nationals had higher PCS and MCS, compared to respondents with other nationalities. Similarly, students without a migration background showed a higher quality of life in PCS and MCS than students with a migration background.

**Table 1. Sociodemographic details of the study population.**

| | | N = 875 | |
|---|---|---|---|
| | | **n** | **%** |
| **Age** | < 20 years | 86 | 9.8 |
| | 20–24 years | 536 | 61.3 |
| | 25–29 years | 169 | 19.3 |
| | > 30 years | 84 | 9.6 |
| | median: 23 | min: 17 | max: 53 |
| **Gender** | Female | 514 | 63.1 |
| | Male | 242 | 29.73 |
| | Non-binary | 38 | 4.7 |
| | Queer | 13 | 1.6 |
| | Ambiguous answer | 7 | 0.9 |
| **German Nationality** | | 837 | 95.7 |
| **Migration background** | | 142 | 16.2 |
| **(Main) subject of study at university** | Medicine | 155 | 17.7 |
| | Law | 138 | 15.8 |
| | Educational Science | 61 | 7.0 |
| | Highschool Teaching | 53 | 6.1 |
| | Sociology | 33 | 3.8 |
| | Elementary School Teaching | 26 | 3.0 |
| | Political Science | 26 | 3.0 |
| | Special Education Teaching | 24 | 2.7 |
| | Pharmacy | 22 | 2.5 |
| | Dental Medicine | 20 | 2.3 |
| | Other | 317 | 36.0 |
| **Aspired degree** | PhD | 39 | 4.5 |
| | State Exam | 496 | 56.7 |
| | University Diploma | 4 | 0.5 |
| | Master | 89 | 10.2 |
| | Bachelor | 240 | 27.4 |
| | Other | 7 | 0.8 |
| **Semesters at university** | 1. semester | 57 | 6.5 |
| | 2–6 semesters | 343 | 39.2 |
| | 7–10 semesters | 283 | 32.3 |
| | 10–15 semesters | 150 | 17.1 |
| | > 15 semesters | 42 | 4.8 |
| | median: 7 | min: 0 | max: 51 |

**Table 2. Stressors reported in the survey.**

| | | n | % |
|---|---|---|---|
| **Membership of a potentially discriminated group** | One group | 198 | 22.6 |
| | Two or more groups | 156 | 17.8 |
| **Experience of discrimination at university** | | 251 | 28.7 |
| **Chronic disease** | | 188 | 21.5 |
| **Insufficient income** | | 160 | 18.3 |
| **Own underage children** | | 42 | 4.8 |
| **Disability** | | 41 | 4.7 |

The subject at university was also associated with quality of life, with medical students having the highest PCS followed by students of Law and Educational Science. For MCS, Law students had the highest score followed by medical students and students of Educational Science.

Students' experience of discrimination was also associated with their quality of life, whereby the differences were particularly pronounced for MCS: Students who reported belonging to at least one group potentially affected by discrimination had an MCS of 33.9 compared to 51.0 among students who reported no membership of a potentially discriminated group. Students with multiple group memberships had substantially lower MCSs than those with only one and those with none. Similarly, students who experienced discrimination in the context of university had lower PCS and MCS than those who did not. More details are displayed in Table 3.

**Table 3. Health-related quality of life (measured using SF-12).**

| | | n | PCS | | MCS | |
|---|---|---|---|---|---|---|
| | | | mean | 95%CI | mean | 95%CI |
| **Total** | | 875 | 51.15 | 47.07–55.23 | 36.34 | 29.77–42.91 |
| **Gender** | Female | 581 | 51.25 | 45.51–56.99 | 35.1 | 26.97–43.23 |
| | Male | 271 | 50.95 | 45.25–56.65 | 38.57 | 26.38–50.76 |
| **Nationality** | German | 837 | 54.19 | 43.29–65.09 | 48.32 | 36.97–59.67 |
| | Other | 38 | 49.46 | 32.58–66.34 | 29.69 | 10.05–49.33 |
| **Migration background** | Yes | 142 | 50.32 | 33.50–67.14 | 30.92 | 10.89–50.95 |
| | No | 733 | 53.19 | 41.67–64.71 | 49.87 | 39.38–60.36 |
| **Subject at university** | Medicine | 155 | 58.18 | 58.10–58.26 | 44.69 | 37.48–51.90 |
| | Law | 138 | 48.78 | 37.55–60.01 | 48.26 | 32.80–63.72 |
| | Educational Sciences | 61 | 39.44 | 31.17–47.71 | 33.49 | 31.29–35.69 |
| **Intended degree** | State Exam | 496 | 52.74 | 42.51–62.97 | 49.27 | 37.08–61.46 |
| | Bachelor | 240 | 44.77 | 28.91–60.63 | 30.48 | 20.07–40.89 |
| | Master | 89 | 56.36 | 44.95–67.77 | 26.11 | 0.69–51.53 |
| | PhD | 39 | 50.25 | 28.18–72.32 | 42.66 | 29.82–55.50 |
| | University Diploma | 4 | . | . | . | . |
| | Other | 7 | . | . | . | . |
| **Semesters at university** | 1. Semester | 57 | . | . | | . |
| | 2–6 semesters | 343 | 48.89 | 30.03–67.75 | 26.97 | 8.98–44.96 |
| | 7–10 semesters | 283 | 53.78 | 39.28–68.28 | 42.57 | 31.44–53.70 |
| | 10–15 semesters | 150 | 51.27 | 42.31–60.23 | 44.71 | 11.82–77.60 |
| | > 15 semesters | 42 | . | . | . | . |
| **Membership of a potentially discriminated group** | Yes | 373 | 51.59 | 35.22–67.96 | 33.9 | 10.97–56.83 |
| | No | 502 | 48.46 | 43.76–53.16 | 51 | 37.89–64.11 |
| **Number of groups** | 1 | 198 | 51.23 | 33.30–69.16 | 38.11 | 16.51–59.71 |
| | 2 or more | 156 | 51.77 | 34.97–68.57 | 31.79 | 7.96–55.62 |
| **Experience of discrimination** | Yes | 251 | 46.59 | 29.40–63.78 | 29.69 | 16.22–43.16 |
| | No | 624 | 52.97 | 39.23–66.71 | 38.99 | 12.33–65.65 |
| **Sufficient income** | Yes | 715 | 54.89 | 42.13–67.65 | 37.55 | 10.48–64.62 |
| | No | 160 | 46.15 | 32.84–59.46 | 34.74 | 11.69–57.79 |
| **Chronic illness** | Yes | 188 | 48.46 | 43.76–53.16 | 51 | 37.89–64.11 |
| | No | 687 | 51.59 | 35.22–67.96 | 33.89 | 10.96–56.82 |

**Table 4. Regression coefficients in simple and multivariable linear regression of the physical dimension of quality of life (PCS) and its determinants.**

| | Univariable Models | | | Multivariable Model | | |
|---|---|---|---|---|---|---|
| | β | 95%CI | R² | β | 95%CI | R² |
| Subject (Educational Science vs Medicine) | -18.75 | -30.08 – -7.42 | 0.58 | -6.94 | -26.19–12.32 | 0.89 |
| Subject (Law vs Medicine) | -9.41 | -19.21–0.39 | | -9.16 | -19.26–0.94 | |
| Subject (other vs Medicine) | -3.90 | -13.15–5.35 | | -0.48 | -9.78–8.83 | |
| Sufficient financial means (no vs yes) | -8.74 | -15.76 – -1.72 | 0.33 | -4.59 | -10.63–1.44 | |
| Experience of discrimination (yes vs no) | -6.38 | -15.06–2.30 | 0.15 | -0.83 | -11.2–9.54 | |
| Religion (yes vs no) | 6.27 | -1.45–13.99 | 0.17 | 9.52 | -0.66–19.7 | |
| Membership of a potentially discriminated group (yes vs no)* | 3.13 | -8.88–15.14 | 0.02 | . | . | |
| Nationality (other nationality vs German)* | -4.73 | -13.18–3.72 | 0.09 | . | . | |
| Migration background (yes vs no) | -2.87 | -12.14–6.40 | 0.03 | -12.25 | -24.19 – -0.31 | |
| Gender (female vs male) | 0.30 | -8.56–9.16 | 0.0003 | 1.67 | -6.09–9.41 | |

## Regression analysis of determinants of quality of life

In the simple regression analyses, low quality of life in the physical dimension measured by PCS was most strongly associated with studying Educational science and studying Law, with having insufficient financial means and with the experience of discrimination at university.

When adjusting for potential confounding in the multivariable analysis, we saw for PCS that the association with the experience of discrimination and of lacking sufficient financial means decreased, and the association with gender, religion and migration background increased. However, as displayed in Table 4, confidence intervals in the multivariable model were wide and included the null value, therefore interpretation of these findings needs to be cautious.

A poor outcome in the mental dimension of quality of life (MCS) was most strongly associated with having a migration background and a nationality other than German, studying a subject grouped under other (such as sociology, political science or pharmacology), and belonging to a potentially discriminated group.

In the multivariable model, migration background and the academic subject displayed the strongest associations, while the influence of financial means, religion and discrimination experience were inverted from a negative to a positive association with the mental dimension of quality of life. For the influence of insufficient financial means, exploratory stepwise regression showed that adjustment for migration background seems to cause this inversion. Again, confidence intervals in the multivariable model were wide and often included the null value, therefore interpretation of these findings needs to be cautious. The details are displayed in Table 5.

## Discussion

Our study found an overall mean quality of life of about 51 (95%CI: 47; 55) for the physical dimension and 36 (95%CI: 23; 43) for the mental dimension with marked differences between different groups of students. In the following, we will discuss these findings in three steps: First, we will compare our findings during the pandemic with the pre-pandemic literature. Then, we will elaborate on some differences between our findings and previous studies and sketch some potential explanations for these differences. Lastly, we will discuss some hypotheses concerning the importance of migration background in our sample and then draw some conclusions.

**Table 5. Regression coefficients in univariable and multivariable linear regression of the mental dimension of quality of life (MCS) and its determinants.**

| Parameter | Univariable Models | | | Multivariable Model | | |
|---|---|---|---|---|---|---|
| | β | 95%CI | R² | β | 95%CI | R² |
| Subject (Educational Science vs Medicine) | -11.19 | -28.22–5.84 | 0.63 | -11.72 | -31.7–8.27 | 0.95 |
| Subject (Law vs Medicine) | 3.57 | -11.19–18.33 | | -17.67 | -28.15 – -7.18 | |
| Subject (other vs Medicine) | -18.12 | -32.04 – -4.20 | | -24.9 | -34.56–15.24 | |
| Sufficient financial means (no vs yes) | -2.81 | -16.57–10.95 | 0.01 | 6.78 | 0.51–13.04 | |
| Experience of discrimination (yes vs no) | -9.31 | -23.54–4.92 | 0.12 | 1.07 | -9.7–11.83 | |
| Religion (yes vs no) | -9.92 | -22.42–2.58 | 0.17 | 2.28 | -8.29–12.85 | |
| Membership of a potentially discriminated group (yes vs no)* | -17.11 | -34.12 – -0.10 | 0.24 | . | . | |
| Nationality (other nationality vs German)* | -18.63 | -28.29 – -8.97 | 0.54 | . | . | |
| Migration background (yes vs no) | -18.95 | -29.67 – -8.23 | 0.49 | -26.07 | -38.46 – -13.68 | |
| Gender (female vs male) | -3.47 | -17.64–10.70 | 0.02 | -3.31 | -11.35–7.72 | |

## Quality of life before and during the pandemic

When comparing our findings to other studies that measured university students' health-related quality of life, our findings align with the literature in two dimensions: First, a comparatively low quality of life especially in the mental dimension is a consistent finding among university students. These low levels remained consistent before and during the pandemic at a similar level. For instance, German studies among fourth-semester medical students found an MCS of 44 in 2008 [35] and of 38.5 in 2016 [36], with both values being substantially lower than the mean of 53 for a German reference population aged 21 to 30 years [35]. Similarly, a pre-pandemic study among students in different colleges and universities in the US found an MCS of 38, which is about one standard deviation below the mean value in the US adult population [37]. In comparison, an Italian study among medical students measured an overall PCS of 54 and an MCS of 41 during the pandemic [38].

It thus seems that students' health-related quality of life did not differ substantially between the times before and during the pandemic, while at both points in time it was considerably lower than other populations' quality of life.

This low mental health-related quality of life in our population is even more striking when compared to patient populations. While at first glance, we would expect patients suffering from chronic conditions to have a much lower health-related quality of life compared to university students, patients' values are of a similar magnitude or even better. A study among mentally ill patients measured an MCS of 37 [39], while a study among diabetes patients found an MCS of 48 [32] and a meta-analysis of studies on quality of life among chronic heart failure patients found an MCS of 51 [40].

The reasons for students' low quality of life and their high burden of mental health issues are not yet fully understood. Studies already established that they are *not* the result of age or generation effects, since students also have a lower quality of life than non-student peers of the same generation and age [35, 41]. Meanwhile, it seems that students' social integration could partly explain their mental health: Social disconnection, homesickness and separation from parents have been shown to predict mental health problems in students, according to a recent systematic review by Sheldon et al. [42] and other studies [43], and might be more common among university students than among their peers who pursue vocational training or start work.

## Differing results in mental health and quality of life among university students

With respect to the associations of sociodemographic characteristics with quality of life, we see some differences between our study and previous studies. These differences mostly pertain to the mental dimension of quality of life: We found that for the *physical dimension* of quality of life, the confidence intervals of the regression coefficients in the regression models were large and included the null value, which implies a considerable uncertainty in interpreting the findings.

In contrast, for the *mental dimension* of quality of life, the model identified migration background and studying a subject other than Medicine as risk factors associated with impaired mental health. There was no relevant association with gender, religion, and with the experience of discrimination.

These findings partly contradict other studies, which usually find substantially higher quality of life in men compared to women [3, 44, 45] and where medical students often have the lowest quality of life [3, 46]. While the reasons for the comparatively good quality of life among female students in our sample are unclear, the relatively good quality of life among medical students could–at least partly–be explained as an effect of our sampling strategy: Due to the research team's affiliation with the Medical Faculty, it is possible that medical students were more likely to receive an invitation and participate in the study than students from other faculties, which suggests that the medical students participating in the study are more representative of the overall population of medical students. Meanwhile, students of other faculties were mainly reached through student associations and counselling offices, which might have oversampled students who sought help from these institutions and might have a higher risk for mental health issues and lower quality of life.

Differences to the literature are also seen with respect to the importance of financial precarity: In the multivariable model, lack of financial means was associated with a better quality of life in the mental dimension in our sample, which is in contrast to most other studies, which find an association between financial stress and impaired mental health [4]. While in simple regression analysis, insufficient financial means were–as in most other studies–associated with lower MCS, exploratory stepwise regression showed that adjustment for migration background is the reason for the conversion from a negative association in simple regression analysis to a positive association in multivariable analysis. A similar effect is seen when adjusting for nationality.

## The relevance of migration background for mental health

In sum, migration background is not only the most important predictor of low quality of life in university students but also seems to be partially mediated by the influence of financial precarity.

To make sense of this finding, we would like to offer two hypotheses: First, the lack of association between financial situation and mental health after adjustment for migration status might be due to the fact that migrants are often excluded from government student loans (which is reserved for German nationals and certain categories of non-Germans). Resultantly, migrants are more likely to be in a financially insecure situation than students without a migration background. Migration background therefore partly seems to function as a surrogate parameter for financial precarity in our model. This hypothesis is supported by other studies that similarly show that students with a migration background more often report lacking financial means [47].

Beyond that, our analysis points to the importance of migration background for both the mental and physical dimension of quality of life. Hereby, the association of migration background with poor quality of life is stronger in the multivariable analysis than in the simple regression analysis. At the same time, membership of a potentially discriminated group is also associated with a lower quality of life in the simple regression model for MCS. This is in line with many other studies that find effects of minority status or the experience of discrimination for a vast array of health-related outcomes [42, 48, 49], which is usually explained by increased stress-levels stemming from the experience of exclusion and relative deprivation [50] and from the physiological reaction to discrimination [51].

Similarly, many studies also show that the experience of discrimination leads to worse subjective health, lower quality of life, worse mental health and higher mortality [51]; this trend does not strongly resonate in our data. In our study, this association is rather weak and the confidence intervals include the null value. We therefore hypothesize that the association of discrimination experiences with low quality of life is confounded by migration background and thus ceases after adjusting for it.

In addition, we hypothesize that the strong association between migration background and low quality of life is only partially mediated by the experience of discrimination and more strongly influenced by social integration and a sense of belonging. A recent systematic review has shown that a sense of social belonging at university is an important predictor of mental health, and that students with a migration background are more likely to feel like they are not belonging [47]. These findings have recently been confirmed by a study specifically for the context of German university [52]. Correspondingly, the establishment or increase of peer support groups has been shown to be effective for improving mental health among international students in the USA [53] and among Black women studying at predominantly white institutions [54].

But also beyond the situation of student minority groups, studies have shown that social support and social integration at university are important influences [42, 47, 55] and therefore can be used as starting points for interventions to increase university students' resilience e.g. through communal activities and student unions. Also, it has been shown that peer support through mentoring programs is effective in reducing students' stress and facilitating their transition into university [56] while students' mental health can be improved by targeted eHealth interventions, as shown in a recent umbrella review [57].

Since we did not gather any data on students' sense of belonging or their social network, we cannot test this hypothesis but suggest that it will be validated in further studies.

## Strengths and limitations

To our knowledge, this is one of very few recent studies investigating the health-related quality of life of university students across different subjects and faculties in a fairly large sample, while also analyzing associated risk factors under the conditions of the COVID-19 pandemic. Hereby, the use of standard instruments ensured a good data quality and the regression models achieved a high $R^2$.

Still, some limitations need to be considered: First, we were not able to distribute invitations for this survey via the university's mailing list, so that we cannot know how many students were actually reached by our invitation; therefore, we cannot calculate a response proportion. Secondly, our invitation might have reached some faculties' students better than others, so that in our realized sample, not all faculties might be represented in proper proportion. Thirdly, this survey investigated the situation at our own university, which might have prompted students towards socially desirable answers. Fourth, some students might counteract the stress

they experience using stress management strategies [8] or receive professional counselling. Since we did not collect data on students' coping strategies, we cannot rule out that some of the differences we find between groups of students come about by differences in the use of stress coping techniques.

## Conclusions

University students are a population group in a vulnerable phase of their biography. Consequently, they are prone to experience stress, which affects their health-related quality of life, especially in the mental health dimension. In line with previous findings, our study found an overall low quality of life during the COVID-19 pandemic, which is particularly pronounced in the mental dimension. Hereby, some groups of students fared even worse than the average, whereby migration background showed the strongest association with poor quality of life after adjustment for other influences. Thus, our research highlights the importance of recognizing differences in students' quality of life and the need for customized support to help marginalized groups. Universities could use these findings to improve their students' well-being in different ways: First, our findings could inform measures that take the unequal distribution of quality of life into account and specifically address particularly burdened groups of students. Second, universities could build on our findings by generally striving for more salutogenic campuses to increase the overall quality of life among their students. Hereby, peer support systems seem to be a promising approach and have been proven helpful in various settings. Despite these approaches to improve university students' health-related quality of life, the overall reasons for students' low quality of life and especially their comparatively poor mental health are still unclear. Here, further research is needed to identify causal pathways.

## Author Contributions

**Conceptualization:** Karoline Wagner, Zoë Reinhardt, Lena Weber, Amand Führer.

**Data curation:** Karoline Wagner.

**Formal analysis:** Amand Führer.

**Funding acquisition:** Karoline Wagner, Amand Führer.

**Investigation:** Karoline Wagner, Amand Führer.

**Methodology:** Karoline Wagner, Zoë Reinhardt, Andreas Wienke, Amand Führer.

**Project administration:** Karoline Wagner, Amand Führer.

**Software:** Lena Weber.

**Supervision:** Andreas Wienke, Rafael Mikolajczyk.

**Writing – original draft:** Amand Führer.

**Writing – review & editing:** Karoline Wagner, Zoë Reinhardt, Sarah Negash, Lena Weber, Andreas Wienke, Rafael Mikolajczyk.

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
