## [Decision Letter · Decision Letter 0]

11 Mar 2024

PONE-D-23-38902Healthy minds in healthy bodies? University students’ health-related quality of life and its determinants. Results from a cross-sectional survey during the COVID-19 pandemic.PLOS ONE

Dear Dr. Führer,

Thank you for submitting your manuscript to PLOS ONE. After careful consideration, we feel that it has merit but does not fully meet PLOS ONE’s publication criteria as it currently stands. Therefore, we invite you to submit a revised version of the manuscript that addresses the points raised during the review process.

We look forward to receiving your revised manuscript.

Kind regards,

D. Daniel, Ph.D.

Academic Editor

PLOS ONE

Journal Requirements:

This research was supported by a grant from Amadeu Antonio Foundation (grant number 21/003).

3. In the online submission form, you indicated that Data are available from the corresponding author upon reasonable request.

Reviewers' comments:

Reviewer's Responses to Questions

**Comments to the Author**

1. Is the manuscript technically sound, and do the data support the conclusions?

Reviewer #1: Yes

Reviewer #2: Partly

2. Has the statistical analysis been performed appropriately and rigorously? 

Reviewer #1: Yes

Reviewer #2: No

3. Have the authors made all data underlying the findings in their manuscript fully available?

Reviewer #1: Yes

Reviewer #2: No

4. Is the manuscript presented in an intelligible fashion and written in standard English?

Reviewer #1: Yes

Reviewer #2: Yes

5. Review Comments to the Author

**Reviewer #1**: Abstract: To enhance the abstract, a brief background is added in the first paragraph, setting the context for the investigation. The conclusion is recommended to be refined based on specific results, as the current conclusion appears too general. Furthermore, it is suggested to include implications of the research in the last paragraph to provide readers with a clearer understanding of the broader significance of the study.

Introduction: The overall structure of the introduction is well-organized, but it is advised to include a definition of the quality of life (QoL) of students to provide a better understanding for readers. Additionally, in lines 87–89, the impact of COVID-19 on the QoL of students should be clearly stated, supported by information from previous reports or research. Lines 86–87 could benefit from further clarification on the purpose and benefits of the research to strengthen the research gap. Lastly, the rationale behind focusing on the physical and mental aspects of QoL and the choice of SF-12 as the study instrument should be explained.

Sampling: The sampling methodology should be clearly stated, including whether it involves purposive sampling, snowball sampling, or another method. In lines 105–107, it is suggested to calculate the sample size based on the university population, considering the proportion of students from each subject or degree.

Discussion: The discussion section requires improvement by providing more in-depth insights into the results, relating them to existing theories, and considering the context of the university. Specific suggestions include explaining the gender differences in the MCS (result in line 161 - 162), the reasons behind the QoL differences between students with and without migration backgrounds (result in line 163 - 164), and the factors contributing to variations in PCS and MCS scores among different majors (Line 165 – 167). Additionally, explanations for the impact of the pandemic on student health-related QoL (Line 211 – 213), as well as any difference with previous research, should be addressed.

Furthermore, it is recommended to discuss the implications of the research, particularly in terms of how the findings can be applied to enhance the physical and mental well-being of university students.

Conclusion: The conclusion's highlight is deemed too general, and it is suggested to refine it based on specific results, the research context, and the implications of the study. This will provide a stronger and more insightful conclusion that resonates with the study's objectives and contributions.

**Reviewer #2:** 1. Introduction:

- Please specify the social concept and health related quality of life measures in this study and specify the importance of such factors in the existence or lifestyle of university students during the COVID-19 pandemic in Germany, such as Social capital theory or Social determinants of health, etc.

-This is because the study was conducted at only one university in Germany. Therefore, the importance and important characteristics of society and the quality of life of university students should be specified.

2. Samling and statistical analysis:

- This is because sampling cannot be done and does not show the size of the population and determining the proportion of sampling. This may affect the distribution of the data and the data does not come from random variables. Therefore, this may cause limitations in the choice of statistics as specified in the study because they are not consistent with the basic conventions of regression statistics. Authors should adjust the methods of using statistics to analyze relationships appropriately.

3. Results: -

4. Discussion:

- Discussion should be based on objectives. Analysis of the relationship of primary factors to quality of life and explain and support how the discovered factors relate to the quality of life of students during the outbreak. How do social factors in the outbreak support or influence the personal and social characteristics of the participants?

- The overall analysis of the study's findings did not reveal a clear alignment with the initial objectives, particularly regarding the outbreak situation and the psychological well-being of the participants.

Despite the inconclusive findings, this study highlights the potential benefits for vulnerable student groups. These students warrant particular attention and support, especially during disaster or epidemic situations, as social and personal limitations can significantly impact their quality of life, both mentally and socially. This in turn can hinder effective learning and vocational skill development.

6. PLOS authors have the option to publish the peer review history of their article (what does this mean?). If published, this will include your full peer review and any attached files.

Reviewer #1: No

Reviewer #2: No

---

## [Author Response · Author response to Decision Letter 0]

8 Apr 2024

Dear Sir or Madam,

We thank the editor for the opportunity to revise our article and we thank the reviewers for their valuable comments and suggestions. Overall, we feel that the revision further improved the article and we hope that our revisions meet the reviewers’ concerns.

Below, we comment in detail how we implemented each suggestion:

Reviewer 1:

1. Title: “You need to revise the title”

We removed the first part of the title to make it shorter and more concise.

2. Abstract: “Please describe little bit about sociodemographic”

Thank you for this suggestion, which greatly improves the abstract! We added the most important sociodemographic information to the abstract (as much as we could, respecting the word limit of 300 words).

3. Introduction: “From the beginning of paragraph, you should focus on mental issues in Covid 19 pandemic. The explanation about the background of the study looked biased and too wide.”

Thank you for this feedback. We added more information on the relevance of the pandemic and its effects on university students, and hope that this clarifies the background of our research question. 

4. Method: “Sampling. You should explain the sampling method because I can’t see it in the manuscript.”

The explanations concerning our sampling procedure are outlined on lines 117 to 124. Initially, we aimed to construct a sampling frame from the university’s list of students’ email addresses and employ an address-based proportionate sampling taking the respondents’ subject into account. 

Since the university denied us access to this list, we had to employ the strategy described in the manuscript, where we tried to invite as many students as possible using mailing lists and student organizations to distribute invitations. The resulting recruitment strategy is probably best described as “convenience sampling”. We added this information to the manuscript. 

5. Method: “Questionnaire. QoL SF 12, which language does this original questionnaire use? If in English, you should do translation to Germany language then do pilot study again to get the Cronbach alpha.”

We used the German version of the SF-12, which has been validated already in 1998 (Gandek et al. 1998) and has been extensively used since then (see also Morfeld et al. 2011). Therefore, a separate validation study was not needed. We added this information to the manuscript (lines 128-131). 

6. Results: “Why did you not do analysis bivariate between sociodemographic and the QoL Score? Because I do not see any meaning here, you just presented the scoring result and we do not know the correlation with the sociodemographic because you should know to have mental problem, a lot of contributing factors for that.” 

We agree that it is interesting to see how QoL is associated with particular sociodemographic characteristics and that bivariate analyses would have been one way to look at this question. 

Meanwhile, the stratified analyses that we conducted and described in lines 181 to 194 are another way of answering this question (and in some ways more suitable than bivariate regression analyses, because the presentation of mean values for different sociodemographic subgroups allows for better comparison with the literature, where the reporting of mean values for the SF-12 dimensions is very common). 

Hereby, we find that women, non-Germans, migrants and law students have lower QoL than their respective comparison group and the sample’s average. In addition, we present the stratified analyses in Table 3.

We therefore believe that the manuscript and Table 3 already answer the question how sociodemographics and QoL are associated. Nevertheless, if we misunderstood this comment and it aims at something else, we are happy to revisit this question again. 

Reviewer 2:

1. Abstract: “To enhance the abstract, a brief background is added in the first paragraph, setting the context for the investigation. The conclusion is recommended to be refined based on specific results, as the current conclusion appears too general. Furthermore, it is suggested to include implications of the research in the last paragraph to provide readers with a clearer understanding of the broader significance of the study.”

Thank you for these suggestions to improve the abstract. We added an introduction at the beginning of the abstract and revised the conclusion so that it is more closely connected with our findings. Together with the added information on the demographics of the sample (in reaction to reviewer 1), we now utilize the word limit of 300 words and therefore cannot elaborate more on the study’s implications, even though we agree that this would have been beneficial. 

2. Introduction: “The overall structure of the introduction is well-organized, but it is advised to include a definition of the quality of life (QoL) of students to provide a better understanding for readers. Additionally, in lines 87–89, the impact of COVID-19 on the QoL of students should be clearly stated, supported by information from previous reports or research. Lines 86–87 could benefit from further clarification on the purpose and benefits of the research to strengthen the research gap. Lastly, the rationale behind focusing on the physical and mental aspects of QoL and the choice of SF-12 as the study instrument should be explained.”

Thank you for this positive feedback! We are happy to further improve the introduction and included your suggestions at two different places: 

First, we added an explanation that defines QoL and explains why this is an important proxy for both overall well-being but also predictive for a vast array of medical outcomes. 

Second, we elaborated a bit more on the impact of COVID on the situation of students and briefly draft previous studies. Hereby the research gap and the rational for our study become clearer now.

Lastly, we revised the methods section and included an explanation on the choice of the SF12 and explain that we chose this instrument primarily because it is since many years established in quality of life research and therefore offers the option for comparison with other target groups but also with studies from before the pandemic. 

3. Sampling: “The sampling methodology should be clearly stated, including whether it involves purposive sampling, snowball sampling, or another method. In lines 105–107, it is suggested to calculate the sample size based on the university population, considering the proportion of students from each subject or degree.”

The sampling strategy is – as discussed in the manuscript – one of the study’s limitations: Initially, we aimed to construct a sampling frame from the university’s list of students’ email addresses and employ an address-based proportionate sampling taking the respondents’ subject into account. 

Unfortunately, university denied us access to this list. Therefore, we had to employ the strategy described in the manuscript, where we tried to invite as many students as possible using mailing lists and student organizations to distribute invitations. The resulting recruitment strategy is probably best described as “convenience sampling”. We added this information to the manuscript.

With respect to the second part of the reviewer’s comment, we are not sure if we understand it correctly. Our approach is exploratory in nature (as opposed to confirmatory), therefore we chose not to perform any sample size calculations beforehand. In our view, this is in line with current guidelines and established practice (e.g. Wasserstein and Lazar 2016 or Greenland et al. 2016). In case we misunderstood the reviewer’s suggestion, we are happy to comply with it after further explanation. 

4. Discussion: “The discussion section requires improvement by providing more in-depth insights into the results, relating them to existing theories, and considering the context of the university. Specific suggestions include explaining the gender differences in the MCS (result in line 161 - 162), the reasons behind the QoL differences between students with and without migration backgrounds (result in line 163 - 164), and the factors contributing to variations in PCS and MCS scores among different majors (Line 165 – 167). Additionally, explanations for the impact of the pandemic on student health-related QoL (Line 211 – 213), as well as any difference with previous research, should be addressed.” 

Thank you for these detailed suggestions for improvements of the discussion section. We are happy to take this opportunity to increase the clarity and rigor of the discussion. To that end, we now structure the discussion by using subheadings and elaborate on each of the topics in a separate paragraph. In addition, we added more references to previous studies and give more space to theoretical explanations of our findings, especially with respect to the importance of migration background as a predictor of quality of life.

5. “Furthermore, it is recommended to discuss the implications of the research, particularly in terms of how the findings can be applied to enhance the physical and mental well-being of university students.” 

AND 

“The conclusion's highlight is deemed too general, and it is suggested to refine it based on specific results, the research context, and the implications of the study. This will provide a stronger and more insightful conclusion that resonates with the study's objectives and contributions.”

Thank you for this suggestion. We agree that the conclusion was too general. We revised it and hereby tried to connect it more closely with our findings and also be more specific in terms of practical implications.

---

## [Decision Letter · Decision Letter 1]

18 Jun 2024

PONE-D-23-38902R1University students’ health-related quality of life and its determinants. Results from a cross-sectional survey during the COVID-19 pandemic.PLOS ONE

Dear Dr. Führer,

Thank you for submitting your manuscript to PLOS ONE. After careful consideration, we feel that it has merit but does not fully meet PLOS ONE’s publication criteria as it currently stands. Therefore, we invite you to submit a revised version of the manuscript that addresses the points raised during the review process.

**ACADEMIC EDITOR: **Please see some minor comments from the reviewer and update your draft accordingly. Thank you

We look forward to receiving your revised manuscript.

Kind regards,

D. Daniel, Ph.D.

Academic Editor

PLOS ONE

Journal Requirements:

Reviewers' comments:

Reviewer's Responses to Questions

**Comments to the Author**

1. If the authors have adequately addressed your comments raised in a previous round of review and you feel that this manuscript is now acceptable for publication, you may indicate that here to bypass the “Comments to the Author” section, enter your conflict of interest statement in the “Confidential to Editor” section, and submit your "Accept" recommendation.

Reviewer #1: All comments have been addressed

2. Is the manuscript technically sound, and do the data support the conclusions?

Reviewer #1: Yes

3. Has the statistical analysis been performed appropriately and rigorously? 

Reviewer #1: Yes

4. Have the authors made all data underlying the findings in their manuscript fully available?

Reviewer #1: Yes

5. Is the manuscript presented in an intelligible fashion and written in standard English?

Reviewer #1: Yes

6. Review Comments to the Author

Reviewer #1: Abstract

• Please use consistent decimal formatting throughout the manuscript. For instance, on line 38, change '63% were female' to '63.0% were female,' because most of the decimal formatting in your manuscript follows the pattern of having one decimal point.

Discussion

• Line 257: I recommend changing the topic to something specific and related to your discussion, such as sociodemographic characteristics and university students’ quality of life, because your current topic might not be clear to readers.

Conclusion

• Lines 342 – 354: I recommend linking the implications of the study in the discussion part within the related topic rather than the conclusion part. After that, summarize the implications in the conclusion part without citing them in the text. For example, in Line 342 - 346, Universities could use these findings to improve their students’ well-being in different ways: First, our findings could inform measures that take the unequal distribution of quality of life into account and specifically address particularly burdened groups of students. This could entail the establishment or increase of peer support groups, which has previously been shown to be effective for international students in the USA [53] and for Black women studying at predominantly white institutions [54]. This aspect, which is highlighted in the discussion regarding the relevance of migration background for mental health, should be addressed in the final or a related paragraph of the discussion. Afterward, you can summarize these aspects in the conclusion. For example, in the conclusion, the research highlights the importance of recognizing differences in students' quality of life and the need for customized support to help marginalized groups. One effective solution is strengthening peer support systems, which have proven helpful in various situations. These results underscore the significance of addressing issues such as migration background when discussing mental health among college students.

7. PLOS authors have the option to publish the peer review history of their article (what does this mean?). If published, this will include your full peer review and any attached files.

Reviewer #1: No

---

## [Author Response · Author response to Decision Letter 1]

8 Jul 2024

Dear Sir or Madam,

We thank the editor for the opportunity to revise our article once more and we thank the reviewers for their detailed comments. Overall, we feel that the revision further improved the article and we hope that our revisions meet the reviewers’ concerns.

Below, we comment in detail how we implemented each suggestion:

1. “Please use consistent decimal formatting throughout the manuscript. For instance, on line 38, change '63% were female' to '63.0% were female,' because most of the decimal formatting in your manuscript follows the pattern of having one decimal point.”

We thank the reviewer for this high attention to detail and changed the decimal formatting accordingly. In this process, we also unified the formatting of the SF-12 scores and the regression coefficients. For better readability, these changes were not done in the Track-Change-Mode.

2. “Line 257: I recommend changing the topic to something specific and related to your discussion, such as sociodemographic characteristics and university students’ quality of life, because your current topic might not be clear to readers.”

We are not sure if we understand this recommendation correctly: We understand that the reviewer suggests that this paragraph should focus on the association between sociodemographic characteristics and quality of life, which it already does. Might there be a typo in the line number and this recommendation actually refers to some other paragraph?

In any case, we changed the introductory sentences to this paragraph to better guide readers into this paragraph and hope that this meets the reviewer’s concern. 

3. “Lines 342 – 354: I recommend linking the implications of the study in the discussion part within the related topic rather than the conclusion part. After that, summarize the implications in the conclusion part without citing them in the text. For example, in Line 342 - 346, “Universities could use these findings to improve their students’ well-being in different ways: First, our findings could inform measures that take the unequal distribution of quality of life into account and specifically address particularly burdened groups of students. This could entail the establishment or increase of peer support groups, which has previously been shown to be effective for international students in the USA [53] and for Black women studying at predominantly white institutions [54].” 

This aspect, which is highlighted in the discussion regarding the relevance of migration background for mental health, should be addressed in the final or a related paragraph of the discussion. Afterward, you can summarize these aspects in the conclusion. For example, in the conclusion, the research highlights the importance of recognizing differences in students' quality of life and the need for customized support to help marginalized groups. One effective solution is strengthening peer support systems, which have proven helpful in various situations. These results underscore the significance of addressing issues such as migration background when discussing mental health among college students.”

This recommendation aims to shorten the conclusion and include the reflection of our study’s practical implication in the discussion section. We like this idea and tried to implement it by moving part of the conclusions to the discussion section and thus aim for a closer interconnection between the discussion of the findings and the practical implications drawn from them. Then, the conclusion section only very briefly summarizes the conclusions without going into detail. Hereby, we happily take up the reviewer’s suggestion for a concise wording. We have the impression that these changes made the discussion more comprehensive and the conclusions more precise, and we hope that this revision is in line with the changes that the reviewer had in mind.

---

## [Decision Letter · Decision Letter 2]

9 Aug 2024

PONE-D-23-38902R2University students’ health-related quality of life and its determinants. Results from a cross-sectional survey during the COVID-19 pandemic.PLOS ONE

Dear Dr. Führer,

Thank you for submitting your manuscript to PLOS ONE. After careful consideration, we feel that it has merit but does not fully meet PLOS ONE’s publication criteria as it currently stands. Therefore, we invite you to submit a revised version of the manuscript that addresses the points raised during the review process.

**Please see my comment below. ** Please submit your revised manuscript by Sep 23 2024 11:59PM. If you will need more time than this to complete your revisions, please reply to this message or contact the journal office at plosone@plos.org. Please include the following items when submitting your revised manuscript:A rebuttal letter that responds to each point raised by the academic editor and reviewer(s). You should upload this letter as a separate file labeled 'Response to Reviewers'.A marked-up copy of your manuscript that highlights changes made to the original version. You should upload this as a separate file labeled 'Revised Manuscript with Track Changes'.An unmarked version of your revised paper without tracked changes. You should upload this as a separate file labeled 'Manuscript'.If applicable, we recommend that you deposit your laboratory protocols in protocols.io to enhance the reproducibility of your results. Protocols.io assigns your protocol its own identifier (DOI) so that it can be cited independently in the future. For instructions see: https://journals.plos.org/plosone/s/submission-guidelines#loc-laboratory-protocols. Additionally, PLOS ONE offers an option for publishing peer-reviewed Lab Protocol articles, which describe protocols hosted on protocols.io. Read more information on sharing protocols at https://plos.org/protocols?utm_medium=editorial-email&utm_source=authorletters&utm_campaign=protocols.

We look forward to receiving your revised manuscript.

Kind regards,

D. Daniel, Ph.D.

Academic Editor

PLOS ONE

Journal Requirements:

**Additional Editor Comments:**

Thank you for your revision. I have some comments.

1. I notice that you just added a new author in this phase, please explain the reason behind this.

2. There is a paragraph in the introduction that only consists of one sentence. Please avoid this.

3. Again, confidence intervals in the multivariable model were wide and often included the null value, therefore interpretation of these findings needs to be cautious.  I think this is not a big deal and normal in statistical analysis that you find variables that are not significant.

4. For the regression, could you please add other parameters as well, e.g., significance (p-value), standard error, coefficient, and the R2 (goodness of fit).

5. for the confidence interval, please make it like this "-28.22 - 5.84", i.e., using "-" and not ";"

6. Please make the conclusion in one single paragraph

Reviewers' comments:

Reviewer's Responses to Questions

**Comments to the Author**

1. If the authors have adequately addressed your comments raised in a previous round of review and you feel that this manuscript is now acceptable for publication, you may indicate that here to bypass the “Comments to the Author” section, enter your conflict of interest statement in the “Confidential to Editor” section, and submit your "Accept" recommendation.

Reviewer #1: All comments have been addressed

2. Is the manuscript technically sound, and do the data support the conclusions?

Reviewer #1: Yes

3. Has the statistical analysis been performed appropriately and rigorously? 

Reviewer #1: Yes

4. Have the authors made all data underlying the findings in their manuscript fully available?

Reviewer #1: Yes

5. Is the manuscript presented in an intelligible fashion and written in standard English?

Reviewer #1: Yes

6. Review Comments to the Author

**Reviewer #1:** Thank you for giving me the opportunity to review this interesting manuscript. I believe that the manuscript is highly beneficial for university students. However, I would like to suggest changing the topic of the discussion section in Line 257 from "Differences to previous studies" as I have not encountered this topic phrasing before. You might consider using "Different Results in Mental Health and Quality of Life Among University Students" or another topic that better captures the meaning and relates more closely to the discussion content.

7. PLOS authors have the option to publish the peer review history of their article (what does this mean?). If published, this will include your full peer review and any attached files.

Reviewer #1: No

---

## [Author Response · Author response to Decision Letter 2]

12 Aug 2024

Dear Sir or Madam,

We thank the editor for the opportunity to revise our manuscript and we thank the reviewer for his or her suggestion concerning the discussion. We revised the manuscript based on those suggestions and hope that the text now meets all requirements.

Below, we detail how we incorporated each comment:

Editor’s comments:

1. I notice that you just added a new author in this phase, please explain the reason behind this.

While we were discussing the changes for the last revision in the team, we discovered that one of the contributors was missing from the authors’ list. Lena Weber has been contributing to the conceptualization of the study, looked after the technical implementation of the online questionnaire, and was involved in the review and editing of the manuscript in its early stages. 

I suspect that it has been my mistake to forget her when entering the authors into the editorial manager during submission, since she stopped working in my working group a while ago and was not involved in other articles belonging to the project in the last months; In a way, she dropped from my mental radar. 

Since her contribution qualifies her for authorship, I hope that it is still possible to correct my error.

2. There is a paragraph in the introduction that only consists of one sentence. Please avoid this.

Thank you for this hint. We revised the manuscript and made sure now that there are no paragraphs that are that short.

3. Again, confidence intervals in the multivariable model were wide and often included the null value, therefore interpretation of these findings needs to be cautious.  I think this is not a big deal and normal in statistical analysis that you find variables that are not significant.

We agree that this is a normal occurrence. Still, we know that many readers follow the habit of looking only at the regression coefficients and therefore would assume an association where there probably is none. We would therefore like to keep this cautionary sentence to avoid such a misinterpretation of our data.

4. For the regression, could you please add other parameters as well, e.g., significance (p-value), standard error, coefficient, and the R2 (goodness of fit).

In line with the recommendations of the American Statistical Association, we would opt not to report p-values for this exploratory analysis, but we added the R² as a measure of goodness of fit to tables 4 and 5. With regression coefficients, confidence intervals and R², we think that readers now can interpret the results nicely. Adding standard error would in our view be redundant, since it can be derived from the confidence interval. 

5. For the confidence interval, please make it like this "-28.22 - 5.84", i.e., using "-" and not ";"

We changed the format of the confidence intervals.

6. Please make the conclusion in one single paragraph

Yes, we did.

Reviewer #1: Thank you for giving me the opportunity to review this interesting manuscript. I believe that the manuscript is highly beneficial for university students. However, I would like to suggest changing the topic of the discussion section in Line 257 from "Differences to previous studies" as I have not encountered this topic phrasing before. You might consider using "Different Results in Mental Health and Quality of Life Among University Students" or another topic that better captures the meaning and relates more closely to the discussion content.

We thank the reviewer for this suggestion and adopted it with a slight change: Instead of “different results” we would like to use “differing results”. We hope that this meets the reviewer’s concerns.

---

## [Editor Report · Decision Letter 3]

30 Aug 2024

University students’ health-related quality of life and its determinants. Results from a cross-sectional survey during the COVID-19 pandemic.

PONE-D-23-38902R3

Dear Dr. Führer,

We’re pleased to inform you that your manuscript has been judged scientifically suitable for publication and will be formally accepted for publication once it meets all outstanding technical requirements.

Kind regards,

D. Daniel, Ph.D.

Academic Editor

PLOS ONE
---

## [Editor Report · Acceptance letter]

23 Dec 2024

PONE-D-23-38902R3 

PLOS ONE

Dear Dr. Führer, 

I'm pleased to inform you that your manuscript has been deemed suitable for publication in PLOS ONE. Congratulations! Your manuscript is now being handed over to our production team.

Kind regards, 

on behalf of

Mr D. Daniel 

Academic Editor

PLOS ONE